# Mesoscale Modeling for Predicting Effective Properties and Damage Behavior of Geopolymer Concrete

**DOI:** 10.3390/ma18010088

**Published:** 2024-12-28

**Authors:** Feiyu Shi, Shanshan Cheng, Longyuan Li

**Affiliations:** School of Engineering, Computing and Mathematics, University of Plymouth, Plymouth PL4 8AA, UK; shanshan.cheng@plymouth.ac.uk (S.C.); long-yuan.li@plymouth.ac.uk (L.L.)

**Keywords:** mesoscale modeling, geopolymer concrete, effective property, microplane model

## Abstract

Geopolymer concrete is a sustainable construction material and is considered as a promising alternative to traditional Portland cement concrete. However, there is still not much research on the effective properties and damage behavior of geopolymer concrete with consideration of its heterogeneous characteristics by means of mesoscale models combined with the regularized microplane damage model. Here, in this research, an easy and simpler approach for generating concrete mesoscale models and characterizing the angular characteristics of aggregate particles is presented. After the proposed mesoscale modeling was validated by numerical, experimental and theoretical models, it was employed further to predict the effective properties and damage behavior of geopolymer concrete. The obtained results show that the effective elastic modulus and compressive strength of geopolymer concrete were greatly affected by the volume fractions of aggregate, while no significant influence on Poisson’s ratio was found. The evolution of damage and coalescence of cracks were affected by the volume fractions and spatial distribution of aggregate particles, which resulted in the different failure patterns in the mesoscale model of geopolymer concrete manufactured by different volume ratios of aggregate.

## 1. Introduction

The utilization of green and eco-friendly materials in construction reduces the carbon footprint of civil engineering, which is greatly beneficial to tackling the issues of climate change. Geopolymer concrete (GPC) has been proven to be a sustainable material and have a lower greenhouse gas emission when compared with ordinary Portland-cement-based concrete (OPC) [1]. However, applications of GPC in civil engineering have not yet been as numerous and wide as cement-based concrete. One of the reasons is that the understanding of the material properties and damage behavior of GPC is not as good as that of OPC.

The properties of concrete, e.g., elastic modulus, Poisson’s ratio and strength, are very important engineering data for structural design and scientific research. These properties were usually experimentally measured [2,3,4] for a given mix design of concrete. Experimental measurements provide the most accurate data with consideration of the working environment; however, the drawback is that additional tests need to be carried out even though the constituents of concrete remain unchanged, as the mix design is slightly varied, e.g., change in the volume ratio of aggregate. Damage of concrete implies how energy is consumed by concrete constituents and how the material resists the stress and fails when its stress is over its strength limit. The surface damage pattern of concrete can be well seen, but damage from inside the bulk of constituents of the concrete composite is not easy to observe in the experiments.

To overcome the limitation of the experimental method in the measuring of concrete properties and investigating the damage behavior of concrete, alternative approaches like theoretical analysis and numerical methods have been developed. In the research of theoretical models, Zheng et al. [5] developed an analytical method for predicting the elastic modulus of concrete. Bian et al. [6] estimated the effective elastic modulus of concrete based on a modified micro-mechanics model. Theoretical models were able to relate the effective/homogenized properties of concrete to the properties of its constituents; however, these models neglected the effect of the spatial distribution of aggregate particles. For overcoming such a drawback, techniques of mesoscale modeling were developed to consider the spatial distribution of aggregate and the heterogeneity of concrete. Wriggers et al. [7] proposed a procedure of mesoscale modeling for predicting the homogenized elastic modulus and damage behavior of concrete, which laid a solid foundation for the present research.

Mesoscale modeling is a sophisticated tool for the research of concrete. It has been utilized to investigate the chloride diffusion [8], water transport [9], tensile failure mechanism [10], thermal conductivity [11], and other properties or behavior [12,13,14,15,16] of concrete. Although mesoscale modeling has been extensively adopted in concrete research, no research has been found yet that employs this versatile technique to predict the effective properties and damage behavior of geopolymer concrete. Here, in this contribution at hand, a simpler practice was proposed for the separation/collision checks between aggregate particles in the mesoscale models; besides, the angular characteristic of aggregate was considered by searching for the boundary surface enclosing some random points. Then, mesoscale models generated in this research were combined with a robust regularized microplane damage model to predict the properties and structural response of concrete. The proposed mesoscale modeling is validated by numerical, experimental and theoretical models, and it is shown that the proposed model is able to predict the effective properties and damage behavior of geopolymer concrete.

## 2. Procedure of Mesoscale Modeling

Mesoscale modeling is a numerical technique for solving the governing equilibrium equations of field quantities of interest at the mesoscopic level, which is usually performed by incorporating mesoscale geometric models in the finite element analysis. Thus, the procedure of mesoscale modeling involves three main steps: (1) build up the mesoscale geometric models of geopolymer concrete (GPC); (2) specify the constitutive laws for the constituents of GPC; and (3) apply boundary conditions to mesoscale models and solve for the primary unknowns—usually displacement. Based on the primary unknowns, other field quantities of geopolymer concrete, e.g., stress and strain, can be derived by using kinematic constraints and constitutive laws, which further contributes to the analysis of the effective property and damage behavior of geopolymer concrete.

### 2.1. Mesoscale Geometric Model

Concrete is a composite where aggregate as inclusion is randomly located within the matrix—the mortar. The process of generating concrete mesoscale models involves two main steps: (1) first taking aggregate particles out of a desired gradation curve; (2) randomly placing particles into the mortar domain by means of separation checks. This ‘take-then-place’ process has been well documented in [7] and is illustrated in Figure 1. However, here, in this research, a different technique for separation checks is adopted. This new method for checking the collision between aggregate particles is not only suitable for round-shaped aggregate but also for angular aggregate, which overcomes the limitation of the procedure presented by Wriggers et al. [7]. In addition, here, in this contribution, a simple solution for generating aggregate of an angular shape is also presented.

For performing separation checks among aggregate particles, the generated aggregate particle is first stored in computer memory as a shape matrix, shown in Equation (1) as follows:(1)$$shape\;matrix=\left[\begin{array}{cc} x_{1} & y_{1}\\ x_{2} & y_{2}\\ \vdots & \vdots \\ x_{n} & y_{n} \end{array}\right]$$
where $\left(x_{n},y_{n}\right)$ is the coordinates of aggregate vertices (round-shaped aggregate can be discretized into a 32-polygon). Then, a MATLAB R2024a function named ***polyshape*** (MATLAB functions are all bold italicized in this section) is used to transform the shape matrix into a poly shape object that can be easily manipulated by the functions of computational geometry in MATLAB. After a shape matrix representing an aggregate has been successfully converted into a poly shape object, a buffer around the aggregate can be created by the function ***polybuffer*** to simulate the gap between aggregate particles. Finally, separation checks between aggregate particles can be easily carried out by using function ***isinterior*** to determine whether vertices of one aggregate particle fall within the domain of another aggregate particle.

For generating an angular aggregate particle, some random points following a uniform distribution are first generated within the square domain of length 1 mm as shown in Figure 2. Then, the surface that encloses all the random points is found and this surface is used as the boundary of the angular aggregate particle. Finally, the angular aggregate particle is enlarged by a factor λ based on the principle of area equivalency using Equation (2).
(2)$$\lambda =\sqrt{\frac{\pi {\left(d_{eq}/2\right)}^{2}}{A}}$$
where $d_{eq}$ is the equivalent diameter of aggregate taken from a desired gradation, and *A* is the area of the angular aggregate particle before enlargement.

### 2.2. Damage Behaviors of GPC Constituents

Concrete is, for the sake of simplicity, usually considered as a homogeneous material, and its damage behavior can be characterized by one damage model. However, such an assumption does not reflect the heterogeneous characteristic of concrete. In mesoscale models, mortar and aggregate are treated as independent phases and their behaviors can be described by damage models with different material parameters, which, in one way, is able to capture the heterogeneity of concrete and, in the other way, considers, in nature, the different structural behaviors of mortar and aggregate. Here, in this research, damage behaviors of GPC constituents are characterized by an implicit gradient-regularized microplane damage model proposed by Zreid et al. [17]. Attempts have been made to incorporate different constitutive laws of materials in the mesoscale modeling of concrete as summarized in the review of Wu et al. [18] and Thilakarathna et al. [19]. In the mesoscale modeling of concrete, high stress usually occurs around the interfacial transition zone between the matrix and aggregate. But, due to the large difference in stiffnesses of the aggregate and matrix, high strain usually takes place within the matrix domain, leading to large deformations in matrix elements and causing a difficulty in solution convergence. For lessening the pain in the non-convergence of the solution, many researchers [20,21,22] adopted an explicit solver. However, such attempts would inevitably introduce an inertia effect in the static simulation and would usually be more computationally costly. The present research employs the static Newton–Raphson algorithm, and the mesoscale modeling is carried out in a static condition. To the best knowledge of the authors, this microplane damage model has not been found to be adopted in the mesoscale modeling, but its robustness and efficiency have been verified in some other research studies [17,23]. In the attempted trials of mesoscale modeling carried out by the authors, the regularized microplane damage model also showed the greatest performance in obtaining the converged peak strength and capturing the post-peak behavior of concrete compared to other geo-mechanical material models, e.g., the Menetrey–Willam model and Drucker–Prager model, available in ANSYS 2024 R2.

In the microplane model, a material point is seen as a microsphere that can be further discretized into microplanes. It is assumed that macroscopic free Helmholtz energy $\psi^{mac}$ equals the integral of microscopic free Helmholtz energy $\psi^{mic}$ over all the microplanes [24]. This relation is expressed in Equation (3).
(3)$$\psi^{mac}=\frac{3}{4\pi }\int_{\Omega }\psi^{mic}d\Omega$$

On the microplanes, deviatoric stress vector $\sigma_{D}$ and volumetric stress $\sigma_{V}$ can be calculated by Equations (4) and (5) as follows:(4)$$\sigma_{D}=\frac{\partial \psi^{mic}}{\partial \varepsilon_{D}}=\left(1-d^{mic}\right)G^{mic}\varepsilon_{D}$$
(5)$$\sigma_{V}=\frac{\partial \psi^{mic}}{\partial \varepsilon_{V}}=\left(1-d^{mic}\right)K^{mic}\varepsilon_{V}$$
where $G^{mic}=G$, $K^{mic}=3K$, $d^{mic}$ is the damage variable on microplanes and *K* and *G* are the bulk and shear modulus.

The stress tensor can be calculated by taking the derivative of $\psi^{mac}$ with respect to the strain tensor as in Equation (6):(6)$$\sigma =\frac{\partial \psi^{mac}}{\partial \varepsilon }=\frac{3}{4\pi }\int_{\Omega }\frac{\partial \psi^{mic}}{\partial \varepsilon }d\Omega =\frac{3}{4\pi }\int_{\Omega }\left(V\sigma_{V}+2Dev^{T}\cdot \sigma_{D}\right)d\Omega$$
where $V=\frac{1}{3}\delta$, $Dev=n\cdot \Pi -\frac{1}{3}n\cdot \delta ⊗\delta$, δ is the second-order identity tensor, Π is the symmetric identity tensor of rank 4 and n is the normal to the microplane.

The evolution of the damage variable on the microplane is defined as in Equation (7).
(7)$$d^{mic}=1-\frac{\gamma_{0}^{mic}}{\gamma^{mic}}\left[1-\alpha^{mic}+\alpha^{mic}exp\left(\beta^{mic}\left(\gamma_{0}^{mic}-\gamma^{mic}\right)\right)\right]$$
where $\gamma_{0}^{mic}$ is the damage threshold and $\alpha^{mic}$ and $\beta^{mic}$ are the maximal degradation and the rate of degradation controlling the softening behavior. History variable $\gamma^{mic}$ is defined in Equation (8) as the maximum of the damage threshold $\gamma_{0}^{mic}$ and equivalent strain for each microplane ${\bar{\eta }}^{mic}$.
(8)$$\gamma^{mic}=max\left(\gamma_{0}^{mic},{\bar{\eta }}^{mic}\right)$$

The equivalent strain for each microplane is normalized by Equation (9).
(9)$${\bar{\eta }}^{mic}=\frac{{\bar{\eta }}_{m}}{\eta_{m}}\eta^{mic}$$
where $\eta_{m}$ is the maximum equivalent strain of all microplanes, as in Equation (10).
(10)$$\eta_{m}=\underset{mic=1:21}{\max}\left(\eta^{mic}\right)$$

The equivalent strain is defined as in Equation (11).
(11)$$\eta^{mic}=k_{0}I_{1}+\sqrt{k_{1}^{2}I_{1}^{2}+k_{2}J_{2}}$$
where $k_{0}$, $k_{1}$ and $k_{2}$ are the coefficients, $I_{1}$ is the first invariant of the strain tensor and $J_{2}$ is the second invariant of the deviatoric strain tensor.

Nonlocal equivalent strain ${\bar{\eta }}_{m}$ can be obtained by solving Equation (12):(12)$${\bar{\eta }}_{m}-c\nabla^{2}{\bar{\eta }}_{m}=\eta_{m}\;\nabla {\bar{\eta }}_{m}\cdot n_{b}=0(\mathrm{Neumann}\;\mathrm{boundary}\;\mathrm{condition})$$
where *c* is the nonlocal range parameter.

### 2.3. Boundary Conditions of Mesoscale Modeling

Measurement of the effective properties of concrete and investigation of its damage behavior are usually carried out by means of uniaxial compressive tests. As illustrated in Figure 3, a concrete specimen with width *W* and height *H* was loaded by displacement control in order to capture the softening behavior. Displacement was prescribed on the top edge of the specimen and the bottom edge was fixed in the *Y* direction, except that the degree of freedom of the middle point on the bottom edge was fixed in both *X* and *Y* directions.

The thickness of the interfacial transition zone (ITZ) between the aggregate and mortar was assumed to be zero in this research.

## 3. Validation of Mesoscale Modeling

### 3.1. Mesh Sensitivity

The solution accuracy of finite element analysis is largely affected by the mesh size. A mesh sensitivity analysis was performed using ANSYS here in this contribution at hand to find out the proper mesh size for the follow-up analysis aiming at predicting the effective property and damage behavior of geopolymer concrete. Three mesh sizes—0.5 mm, 1 mm and 1.5 mm—were chosen for the mesoscale model and solutions were compared with the experiment of Cordes [25] and the numerical simulation of Wriggers et al. [7]. According to Ref. [7], the volume fraction of aggregate is 40%, and, based on this information, a mesoscale model as shown in Figure 4 was generated using the proposed procedure. For the sake of simplicity, aggregate particles were treated as spheres in the mesoscale model of Wriggers et al. [7]. Here, in this research, the angular characteristic of aggregate was considered; however, for reducing the computational costs, only 2D geometry was considered. The generated mesoscale concrete specimen of size 100 × 100 mm contains 158 aggregate particles whose sizes range from 2.36 to 19 mm.

Linear material properties of concrete constituents were taken and are shown in Table 1. Strengths of concrete constituents are needed in order to calculate the parameters for the damage model here used by the present research. A general formula [26] as given in Equation (13) relates the compressive strength of concrete to its elastic modulus. This equation can be rearranged as in Equation (14) to estimate the compressive strength of mortar.
(13)$$E_{c}=\alpha {\left(\frac{f_{c}}{k}\right)}^{\beta }+c$$
(14)$$f_{c}=k{\left(\frac{E_{c}-c}{\alpha }\right)}^{1/\beta }$$
where $E_{c}$ and $f_{c}$ are the elastic modulus and compressive strength of mortar and α, β, k and c are coefficients. Here, in this research, these coefficients are taken as α=4700, β=1/2, k=1 and c=0 as suggested in ACI code [27]. For aggregate, the rock type is assumed to be granite and its compressive strength was experimentally measured as 133.2 MPa [28]. Tensile strengths of both mortar and aggregate were estimated using Equation (15) [29].
(15)$$f_{t}=1.4{\left(\frac{f_{c}}{10}\right)}^{2/3}$$

Based on the estimated strengths of mortar and aggregate, parameters for the damage model as shown in Table 2 can be calculated using Equations (16) and (17) [23].
(16)$$k_{0}=k_{1}=\frac{k_{r}-1}{2k_{r}\left(1-2\upsilon \right)}$$
(17)$$k_{2}=\frac{3}{k_{r}{\left(1+\upsilon \right)}^{2}}$$
where $k_{r}$ is the ratio of compressive strength to tensile strength. Other parameters shown in Table 2 for the damage model were taken or estimated from Refs. [17,23].

After determining all the material parameters for the damage models of both mortar and aggregate, mesoscale modeling was performed to simulate the structural behaviors of the concrete specimen with different mesh sizes as shown in Figure 4. A simulation was also carried out to model the behavior of geopolymer concrete in the experiments of Nguyen et al. [30]. The results were compared with experiment and numerical simulation in the work of Wriggers et al. [7] and Nguyen et al. [30] as presented in Figure 5.

The process of parameter calibration is summarized as follows: (1) experimental data: obtain stress–strain data from experiments, e.g., Nguyen et al., for concrete. (2) Initial guess: set the initial empirical damage parameters, Young’s moduli, Poisson’s ratios and other material constants for both aggregate and mortar. (3) Simulation: run an initial simulation of mesoscale modeling using the initial values. (4) Error analysis: compare the FEA results with the experimental stress–strain curve. (5) Adjust parameters: modify material constants. (6) Iterate: repeat until the FEA curve matches the experimental data well.

It can be observed in Figure 5 that the present models captured the initial stiffness, compressive and residual strengths of concrete well. Variations between models with different mesh sizes were negligible and, as the mesh became finer, the compressive strength was much closer to that of the experiment. Thus, the model with a mesh size of 0.5 mm would be used in all the follow-up simulations. Residual strength predicted by the present model was in good agreement with that of the experiment, while the simulation given by Wriggers et al. [7] underestimated it; however, the model in this research overestimated the pre-peak strength and did not capture the strain corresponding to the compressive strength well. This disparity may be partially due to the assumption in this research that mortar elements and aggregate elements were perfectly bonded, and the interfacial transition zone was not considered [7]. The dimension of concrete and shape of aggregate may also contribute to the differences. For the comparison of the stress–strain relationship between the present mesoscale modeling and the experiment of Nguyen et al. [30], a good agreement can be observed except that the geopolymer concrete exhibited slightly more brittleness in the experiment after the residual stress was less than 80% of the peak strength.

### 3.2. Capability for Predicting Effective Property of Concrete

Effective properties of concrete, e.g., elastic modulus, compressive strength and so on, are very important material properties in the structural design and in the research of concrete damage behavior. This section is dedicated to verifying the capability of the proposed mesoscale models combined with the microplane damage model in the prediction of effective properties of concrete. The results will be compared with experimental tests, numerical solutions and the theoretical bounds presented in [7].

Aggregate of type I in Ref. [7] was chosen for the verification. Aggregate sizes range from 2.36 to 12.7 mm. Four mesoscale models with different volume fractions of aggregate, as shown in Figure 6, were generated by the procedure introduced in Section 2.1. Aggregate in all four models was divided into three segments—[2.36, 4.75], [4.75, 9.5], [9.5, 12.7]—exactly as in Ref. [7], and all these four concrete specimens have the size of 100 × 100 mm. The generated mesoscale model with a 20% volume fraction of aggregate contains 81 aggregate particles and the rest of the models contain 116, 165 and 202 aggregate particles, respectively.

Sieve analysis was performed to determine the gradation of the generated aggregate particles in these four models. As presented in Figure 7, aggregate gradations of the generated models are all in good agreement with the desired gradation of the type I aggregate in Ref. [7].

Aggregate and mortar were modeled by the microplane damage model. For the purpose of comparison, elastic material properties of aggregate and mortar were directly taken from Ref. [7] and are listed in Table 3.

Other parameters for the damage model are shown in Table 4. Parameters for the damage function were calculated using the same method as in Section 3.1 and parameter values were rounded to two significant digits. Other parameters were kept unchanged as in Table 2.

Four concrete mesoscale models with different volume fractions of aggregate were subjected to loading conditions as stated in Section 2.3. Uniaxial compression continued until the strain of the top edges of the four models reached 0.05%, and stress was calculated using the engineering definition, which is the reaction force divided by the loading area. Relationships of the stress and strain of these four models are plotted in Figure 8. It can be observed in Figure 8 that the initial stiffness of the concrete specimen was enhanced as the volume fraction of aggregate increased, which means that the higher the volume fraction of aggregate, the higher the effective elastic modulus.

All these four models exhibited linear behavior when the top edges were strained to 0.05%. Here, in this research, elastic moduli were calculated as the slope of the stress–strain curves, and the relationships of effective elastic moduli against aggregate volume fractions are plotted in Figure 9 along with the numerical simulation, experimental test and theoretical bounds presented in [7].

As presented in Figure 9, effective moduli calculated by the mesoscale modeling here in this contribution match almost exactly with those predicted by Wriggers et al. [7]. In the work of Wriggers et al. [7], 3D mesoscale models were used and the angular characteristic of aggregate was not considered. Two-dimensional models with aggregate of angular shape were adopted in the present research. The replicated results imply that, in one way, a more computational cost-effective 2D mesoscale could also be used for predicting effective properties of concrete, and, in the other way, the angular features of aggregate might have little effect on the effective elastic modulus of concrete.

The effective elastic moduli predicted in the present research were also close to those measured by experiments, especially at aggregate volume fractions of 20% and 50%. In addition, the predicted values of effective elastic moduli fell well between the bounds of Hashin and Shtrikman [31,32]. It can be seen in Figure 9 that the predicted data distributed around the lower H-S bound.

At this point, the capability of the mesoscale modeling proposed here in this contribution for predicting the structural behavior and effective property has been verified and validated. This modeling technique will be further applied to predict the effective property of geopolymer concrete (GPC) and to investigate the damage behavior of GPC. The results are to be presented and discussed in the follow-up sections.

## 4. Results and Discussion

### 4.1. Prediction on Effective Properties of Geopolymer Concrete

The packing density of aggregate has significant influence on the strength of geopolymer concrete. The aggregate size that follows Fuller’s gradation [33] as given in Equation (18) usually results in the optimal packing [7]; therefore, Fuller’s curve has been utilized by many researchers [15,20,34,35] in the research of mesoscale modeling. It was also found to be adopted in 2D mesoscale models [36], although Fuller’s curve is much more suitable for 3D mesoscale models.
(18)$$P\left(d\right)={\left(\frac{d}{d_{max}}\right)}^{n}$$
where *P* is the cumulative passing percentage at the sieve opening of *d*, $d_{max}$ is the maximum diameter of aggregate and *n* is an exponent ranging from 0.45 to 0.7.

Based on Fuller’s curve and assuming that *n* = 0.5, Walraven [37] first derived the probability function, as given in Equation (19), of the size distribution of aggregate on a cut-off 2D slice from a 3D specimen.
(19)$$P\left(d\right)=v_{p}\left[a{\left(\frac{d}{d_{max}}\right)}^{0.5}+b{\left(\frac{d}{d_{max}}\right)}^{2}+c{\left(\frac{d}{d_{max}}\right)}^{4}+d{\left(\frac{d}{d_{max}}\right)}^{6}+e{\left(\frac{d}{d_{max}}\right)}^{8}+f{\left(\frac{d}{d_{max}}\right)}^{10}\right]$$
where $v_{p}$ is the volume fraction of aggregate and the coefficients are a=1.455, b=−0.5, c=0.036, d=0.006, e=0.002, f=0.001, respectively. Later, Walraven et al. [38] proposed a modified form as Equation (20).
(20)$$P\left(d\right)=v_{p}\left[a{\left(\frac{d}{d_{max}}\right)}^{0.5}+b{\left(\frac{d}{d_{max}}\right)}^{4}+c{\left(\frac{d}{d_{max}}\right)}^{6}+d{\left(\frac{d}{d_{max}}\right)}^{8}+e{\left(\frac{d}{d_{max}}\right)}^{10}\right]$$
where the coefficients are a=1.065, b=−0.053, c=−0.012, d=−0.0045, e=−0.0025, respectively.

Equation (20) has been extensively adopted to build up the 2D mesoscale models [39,40,41,42]. Here, in this research, it was also applied in the generation of mesoscale models.

The aggregate size ranges from 2.36 to 12.7 mm, and was divided into three segments—[2.36, 4.75], [4.75, 9.5], [9.5, 12.7]. A set of aggregate volume fractions ranging from 20 to 40% was considered, and the generated mesoscale geometric and meshed models are presented in Figure 10 (for better visualization, a mesh of aggregate was not plotted). For convenience of reference, the mesoscale model is denoted as MesoModel_*x*PcAgg, where *x* is the volume fraction. All three concrete specimens have the size of 100 × 100 mm. MesoModel_20PcAgg has 92 aggregate particles while MesoModel_30PcAgg and MesoModel_40PcAgg contain 140 and 186 aggregate particles, respectively.

Geopolymer concrete is manufactured by the replacement of cement with a geopolymer binder and mixing with aggregate. Here, in this research, fly-ash-based geopolymer as a binder and granite as aggregate were considered as the constituents of geopolymer concrete. Experimental tests for measuring the properties of geopolymer [43,44] and granite rock [45] have been reported, and these data will be used for simulations, as listed in Table 5.

Other damage model parameters of geopolymer and granite were calculated and rounded to 2 digits to the right of the decimal point as listed in Table 6.

Mesoscale modeling was performed for these three models. Figure 11 shows the stress–strain relationships of geopolymer concrete with different aggregate volume fractions. As observed from the plot, both the initial stiffness and compressive strength of geopolymer concrete (GPC) were enhanced as the volume fractions of aggregate increased. The residual strength of GPC went up when the volume fraction increased from 20% to 30%; however, it decreased when the volume fraction increased from 30% to 40%. The mechanism behind this phenomenon may be that aggregate strengthened geopolymer at a low volume percentage, but, as the volume ratio increased, there was less geopolymer binding the aggregate together as a whole, causing aggregate sliding after the geopolymer binder failed.

In uniaxial compression, the homogenized elastic modulus of geopolymer concrete, by definition, is equal to the initial slope of the stress–strain curves. Elastic moduli are calculated and plotted along with the linear formulation modified from [46] and Counto’s model [47] against the volume fractions in Figure 12.

Counto’s prediction model [47] and the linear formula [46] by additively combining the elastic moduli of concrete constituents and using their volume fractions as weight factors are expressed in Equations (21) and (22), respectively. It can be clearly observed in Figure 12 that the homogenized elastic moduli predicted by the mesoscale modeling were in quite good agreement with those predicted by Counto’s model [47].
(21)$$\frac{1}{E_{c}}=\frac{1-\sqrt{V_{a}}}{E_{m}}+\frac{1}{\left(\frac{1-\sqrt{V_{a}}}{\sqrt{V_{a}}}\right)E_{m}+E_{a}}$$
(22)$$E_{c}=\left(1-V_{a}\right)E_{m}+V_{a}E_{a}$$
where $E_{c}$ is the effective elastic modulus of concrete, $E_{m}$ and $E_{a}$ are elastic moduli of the binder and aggregate and $V_{a}$ is the volume fraction of aggregate.

The compressive strength of geopolymer concrete is the peak stress on the stress–strain curves in Figure 11. They are quantitatively listed in Table 7. It can be found that the strength of geopolymer concrete increased by about 17% when the volume fraction of aggregate increased from 20% to 30%, while a 37% rise in strength was observed when the volume fraction of aggregate increased from 20% to 40%.

When the material is strained in the axial direction, it would also expand or shrink in its lateral direction. This is Poisson’s effect as depicted in Figure 13. By the definition of engineering strain measurement, axial strain and lateral strain are calculated by Equations (23) and (24) as follows:(23)$$\varepsilon_{axial}=\frac{u_{y}}{H}$$
(24)$$\varepsilon_{lateral}=\frac{{\bar{u}}_{rx}+{\bar{u}}_{lx}}{W}$$
where $u_{y}$ is the axial deformation and ${\bar{u}}_{rx}$ and ${\bar{u}}_{lx}$ are the average deformation of the right and left edges of geopolymer concrete, respectively. H and W are the original height and width of the concrete specimen.

Stress against the axial and lateral strains is plotted in Figure 14. The initial linear deformation of geopolymer concrete and Poisson’s ratio was calculated using Equation (25) and rounded to the third digit to the right of the decimal point, listed in Table 8. It was noticed that the volume fraction had little effect on the Poisson’s ratio of geopolymer concrete.
(25)$$\upsilon =-\frac{\varepsilon_{lateral}}{\varepsilon_{axial}}$$

### 4.2. Damage Behavior of Geopolymer Concrete

The evolution of damage, in one way, indicates a possible region where the initiation, propagation and distribution of cracks takes place; in the other way, it implies the softening and failure of the material. In the microplane damage model, concrete damage is quantified as a damage index ranging from 0 to 1, where 0 means no damage and 1 dictates that the element has been fully damaged.

Figure 15 depicts the damage evolution of geopolymer concrete with different volume fractions of aggregate and the experimental failure modes of geopolymer concrete. It can be seen that damage was first initiated in geopolymer near the upper edge region, where controlled displacement was prescribed. This may be due to the concentration of force causing a large deformation around the upper edge region. Damage then further spread out into the bulk of geopolymer as the loading was increased. With more and more geopolymer elements being damaged, geopolymer concrete exhibited softening behavior and, at the end of the loading, most of the geopolymer was crushed while some of the aggregate remained intact. The spatial distribution and volume fraction of aggregate affected the propagation of damage, and geopolymer concrete with different volume fractions showed different failure patterns.

MesoModel_40PcAgg was taken as an example for further discussion about how geopolymer concrete was damaged and crushed as displacement loading increased. The stress–strain relation and equivalent strain of MesoModel_40PcAgg are presented in Figure 16. At the initial stage, high equivalent strain was first detected in the geopolymer. As the displacement increased, more and more geopolymer elements showed high equivalent strain, causing the damage evolution of geopolymer elements. At the end of the loading, most of the geopolymer binder was damaged and crushed, and equivalent strain accumulated around a V-shaped region at the bottom edge of the concrete.

Figure 17 and Figure 18 present the equivalent stress–strain contours of two constituents of concrete MesoModel_40PcAgg at the peak strength and the evolution of the averaged equivalent stress of all elements against the strain loading, respectively. It can be observed that aggregate exhibited much higher values of stress compared with the geopolymer matrix, which means that the main constituent in concrete for resisting the external loading was aggregate and that the contribution of geopolymer was to hold aggregate particles strongly together. Due to its higher stiffness, the strain of aggregate particles was much smaller than that of the geopolymer matrix. From Figure 18, it can be clearly seen that the stress evolution curve of concrete lay between those of aggregate and mortar, where the stress curve of aggregate was the upper bound and that of the geopolymer matrix was the lower bound. This, for one point, supports the claim that aggregate was the main phase resisting the external loading; for the other point, the geopolymer matrix was strengthened by aggregate particles, which was the mechanism of concrete resisting the external loadings.

A linear combination model similar to Equation (22) was proposed to predict the averaged equivalent stress of geopolymer concrete as given in Equation (26). As shown in Figure 18, the averaged equivalent stress calculated by Equation (26) matches well with the simulated result of concrete.
(26)$$\sigma_{c}=\left(1-V_{a}\right)\sigma_{m}+V_{a}\sigma_{a}$$
where σ is the averaged equivalent stress and subscript *c*, *m* and *a* denote concrete, geopolymer matrix and aggregate, respectively.

## 5. Conclusions

Little research on investigating the effective properties and damage of geopolymer concrete using mesoscale modeling has been found. Here, in this contribution at hand, a simpler procedure for generating mesoscale models of geopolymer concrete with angular-shaped aggregate particles (similar to the realistic aggregate) was presented. Behaviors of the binder and aggregate were modeled by a regularized microplane damage model. After the procedure of mesoscale modeling was calibrated and validated with numerical and experimental models, the effective properties and damage behavior of geopolymer concrete were further investigated, and some of the conclusions drawn from the results are presented as follows:

The presented procedure for generating the mesoscale models of concrete took a simpler approach to building up the angular characteristic of aggregate by searching the boundary enclosing some random points, and this procedure also made the best out of the functions of computational geometry in MATLAB, which contributed to a less laborious process in performing the separation/collision checks among aggregate particles. The generated mesoscale models combined with the microplane damage model had the capability to capture the structural behavior of concrete well and predict the effective properties of concrete well, which has been verified by numerical, experimental and theoretical models.

In mesoscale modeling, due to the different stiffness of the matrix and inclusion, high stress and strain were usually first detected in the elements around the interfacial transition zone between the matrix and inclusion, which sometimes caused distortion of these elements and non-convergence of simulation. The regularized microplane model adopted a non-local approach and was versatile in overcoming the non-convergence of the numerical solution, which has also been validated by the presented research.

The mesoscale modeling presented in this research can accurately predict the effective properties of geopolymer concrete, e.g., homogenized elastic modulus, overall compressive strength, and Poisson’s ratio. The effective elastic modulus predicted by the present research was in good agreement with that predicted by Counto’s formula [47]. The effective elastic modulus and compressive strength of geopolymer concrete were greatly affected by the volume fractions of aggregate, while little effect on Poisson’s ratio was found.

Damage in geopolymer was first initiated, and, as the load increased, more and more geopolymer material was damaged. At the end of the loading, most of the geopolymer was crushed while some of the aggregate remained intact and other aggregate particles were slightly damaged. Volume fractions of aggregate and the spatial distribution of aggregate particles affected the propagation of damage and the coalescence of cracks, which caused different failure patterns of geopolymer concrete with various volume ratios of aggregate.

Little experimental research on the interfacial transition zone (ITZ) of geopolymer concrete was found, and thus the influence of the ITZ was not considered in this research due to the lack of data for material properties of the ITZ. It will be incorporated into the mesoscale models in the future research when more data of the ITZ are available. In addition, we would like to highlight that the results generated from the 2D mesoscale model in our study match well with the experimental data, demonstrating the reliability of this approach for the scope of our current research. However, we recognize the value of a 3D model in capturing additional spatial and vertical dynamics that a 2D model cannot fully resolve. We plan to consider the implementation of a 3D mesoscale model in future research to build on the findings from this study and explore more complex interactions.

## Figures and Tables

**Figure 1 materials-18-00088-f001:**
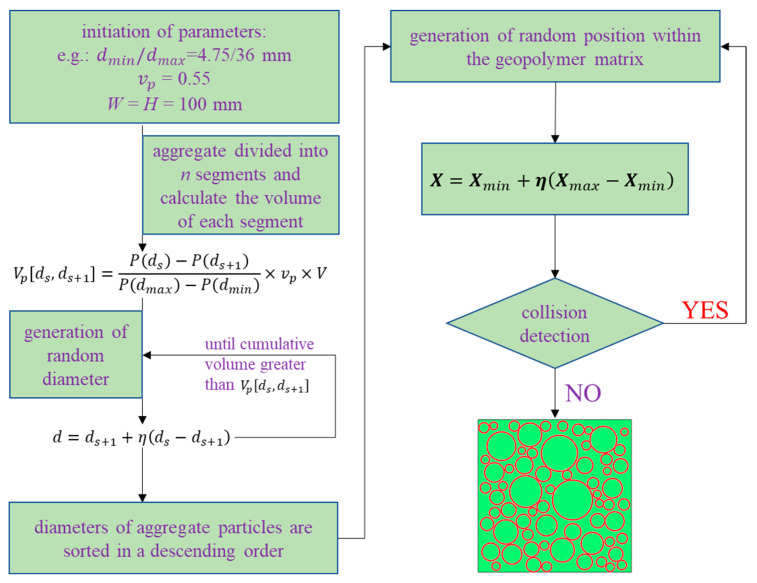
Generative process of mesoscale models.

**Figure 2 materials-18-00088-f002:**
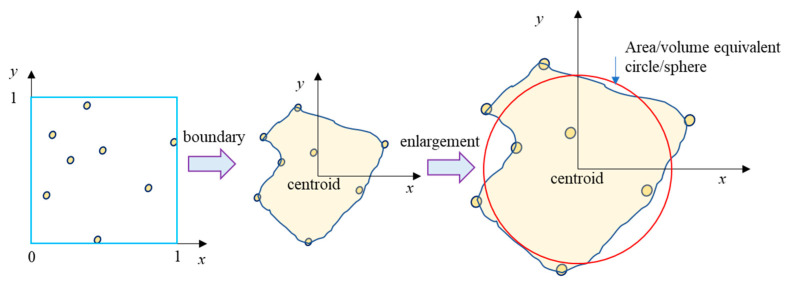
Generation of angular aggregate particle.

**Figure 3 materials-18-00088-f003:**
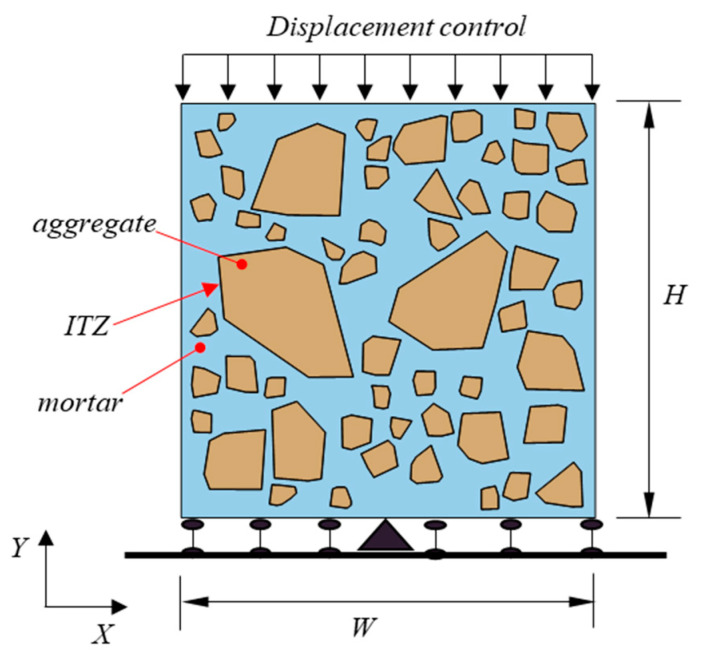
Boundary conditions of mesoscale modeling.

**Figure 4 materials-18-00088-f004:**
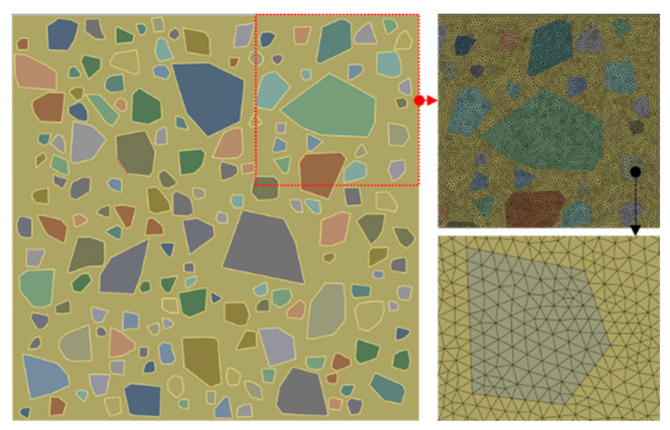
Geometry and finite element mesh of mesoscale model.

**Figure 5 materials-18-00088-f005:**
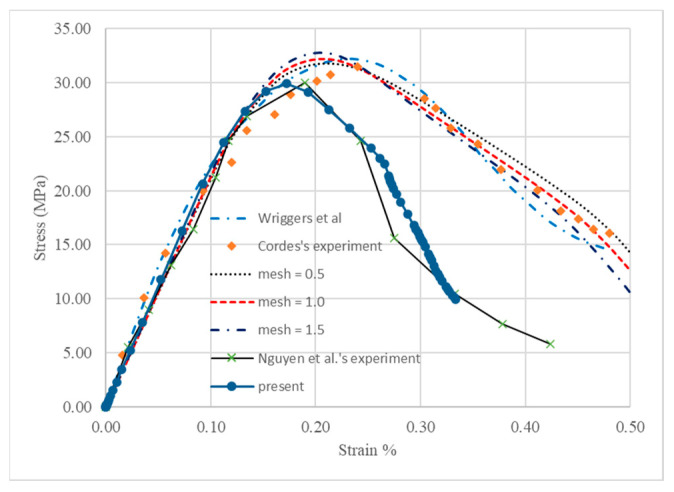
Comparisons of stress–strain behavior of models with different meshes with results of Wriggers et al. [7] and Nguyen et al. [30].

**Figure 6 materials-18-00088-f006:**
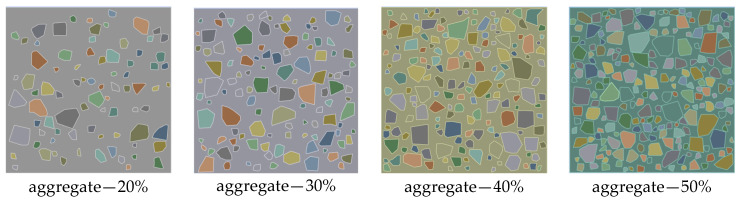
Mesoscale models with different volume fractions of aggregate.

**Figure 7 materials-18-00088-f007:**
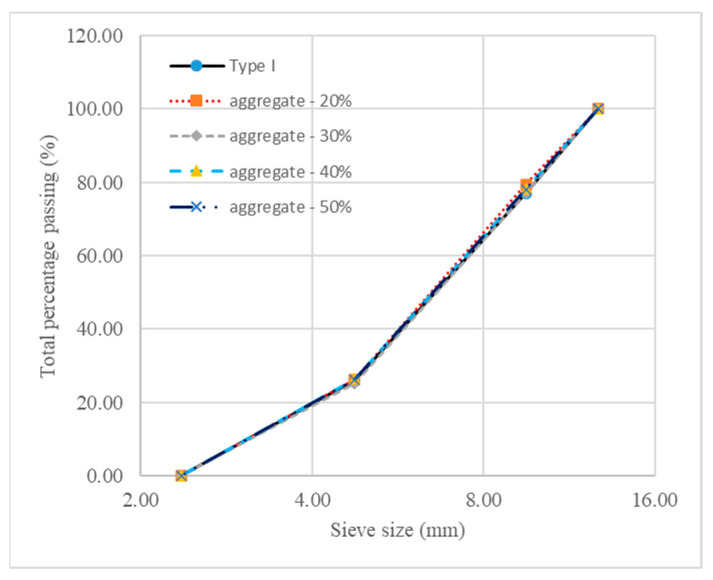
Gradation curves of aggregate particles in different models.

**Figure 8 materials-18-00088-f008:**
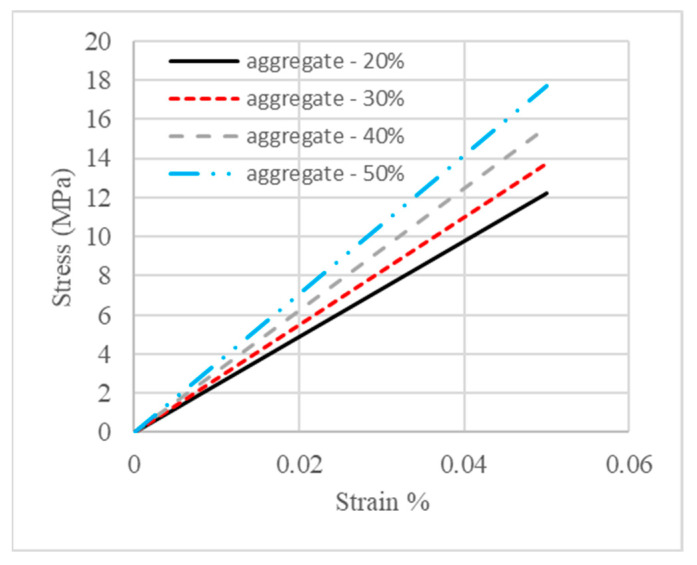
Stress–strain relations of concrete with different aggregate volume ratios.

**Figure 9 materials-18-00088-f009:**
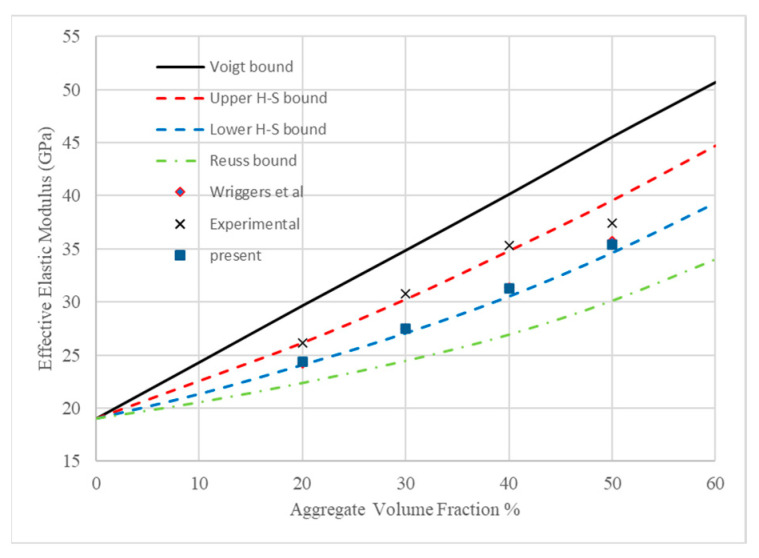
Relationships of effective elastic moduli against aggregate volume fractions and results from Wriggers et al. [7].

**Figure 10 materials-18-00088-f010:**
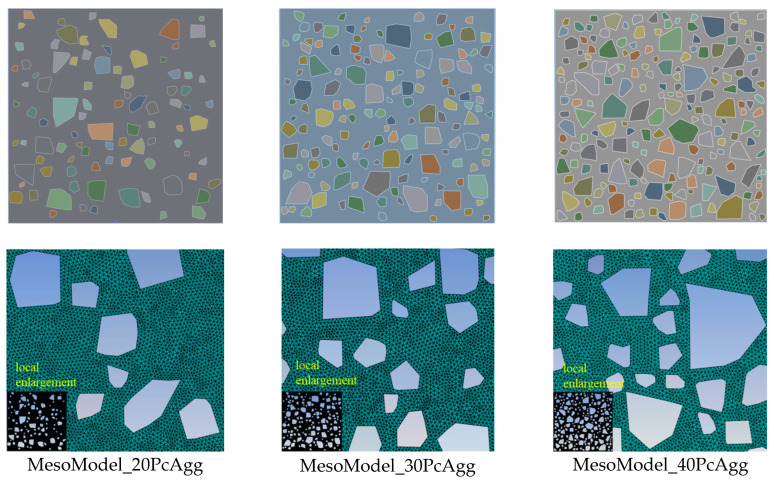
Mesoscale geometric and meshed models with different volume fractions of aggregate.

**Figure 11 materials-18-00088-f011:**
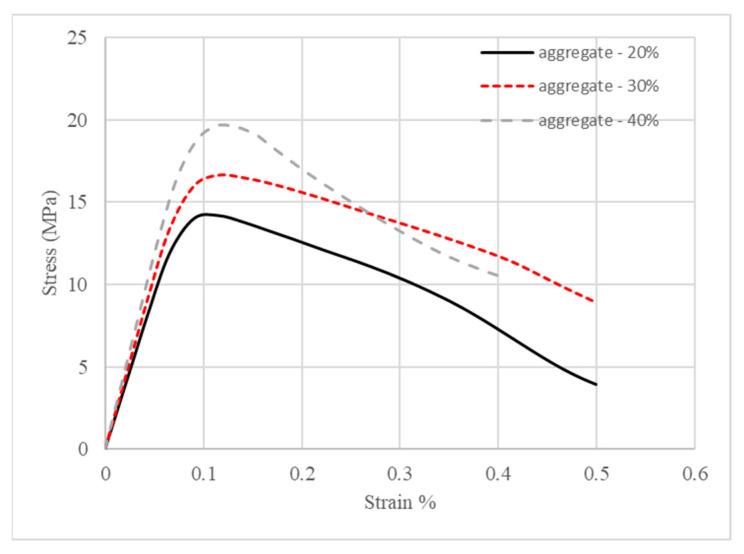
Stress–strain relationships of geopolymer concrete.

**Figure 12 materials-18-00088-f012:**
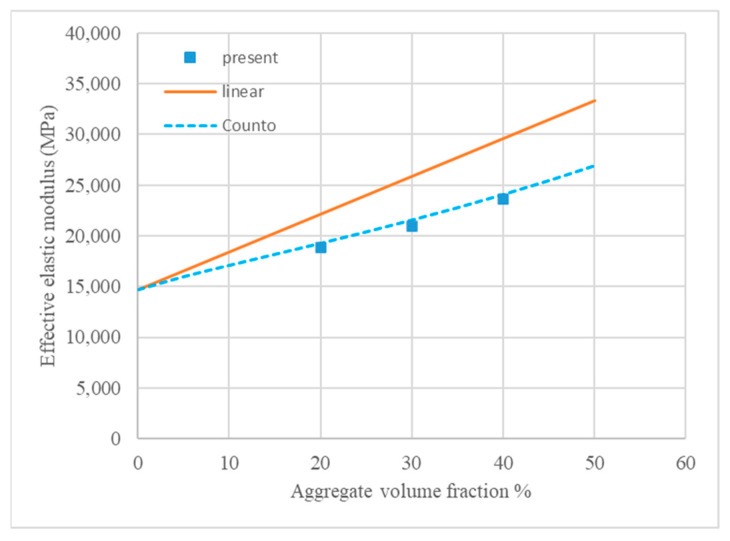
Effective elastic modulus of geopolymer concrete.

**Figure 13 materials-18-00088-f013:**
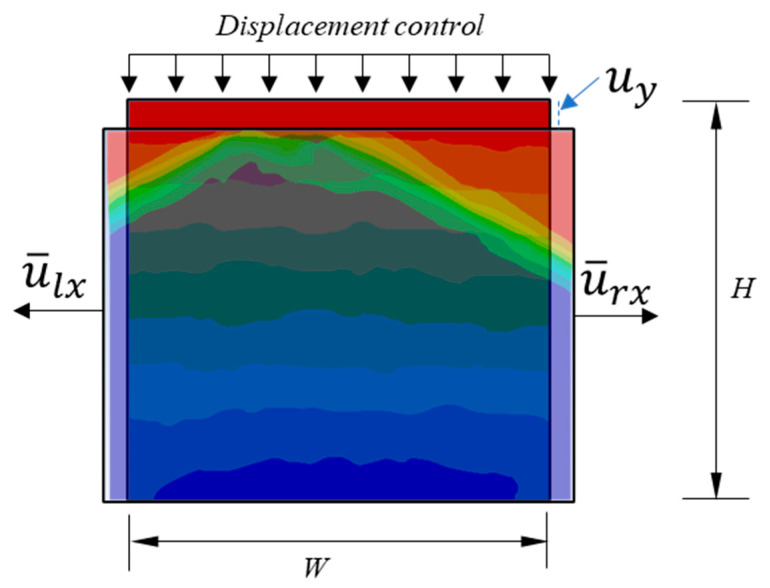
Schematic for Poisson’s effect.

**Figure 14 materials-18-00088-f014:**
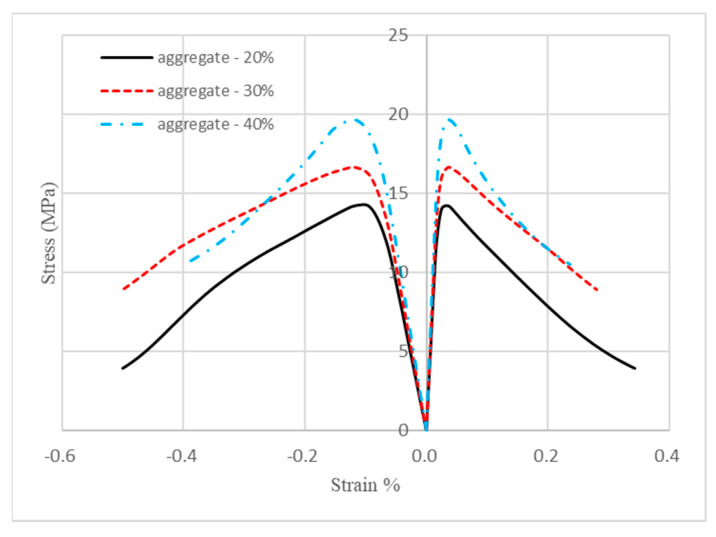
Stress response against axial (**left**) and lateral (**right**) strains.

**Figure 15 materials-18-00088-f015:**
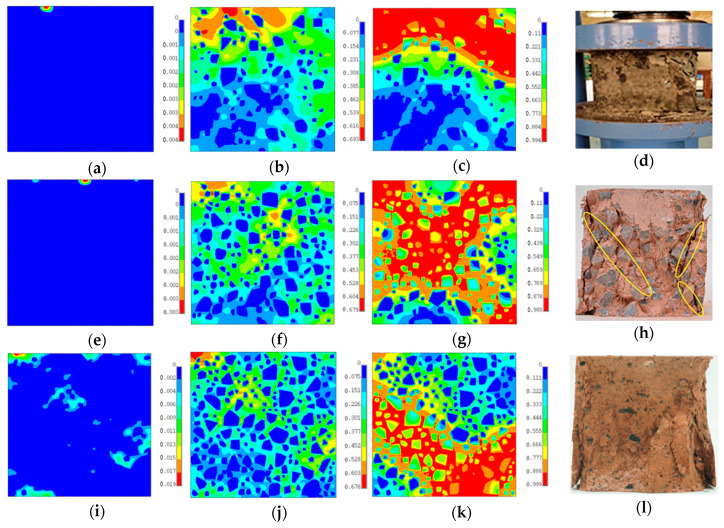
Damage evolution of geopolymer concrete with different volume fractions of aggregate. (**a**) damage initiation, MesoModel_20PcAgg; (**b**) at peak strength, MesoModel_20PcAgg; (**c**) at residual strength, MesoModel_20PcAgg; (**d**) Kanagaraj et al., 2022, in Ref. [48]; (**e**) damage initiation, MesoModel_30PcAgg; (**f**) at peak strength, MesoModel_30PcAgg; (**g**) at residual strength, MesoModel_30PcAgg; (**h**) Zhong et al., 2023, in Ref. [49]; (**i**) damage initiation, MesoModel_40PcAgg; (**j**) at peak strength, MesoModel_40PcAgg; (**k**) at residual strength, MesoModel_40PcAgg; (**l**) Fan et al., 2023, in Ref. [50].

**Figure 16 materials-18-00088-f016:**
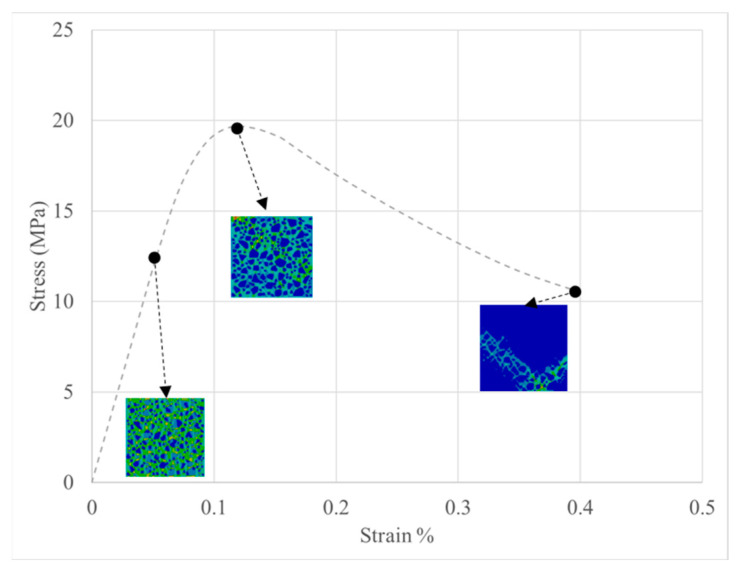
Stress response and equivalent strain of MesoModel_40PcAgg.

**Figure 17 materials-18-00088-f017:**
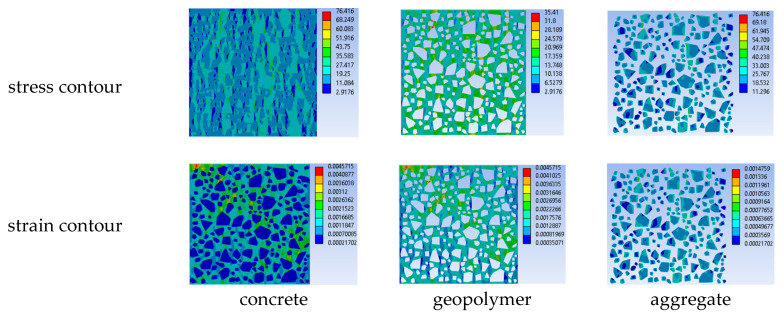
Equivalent stress–strain contours at peak strength of MesoModel_40PcAgg.

**Figure 18 materials-18-00088-f018:**
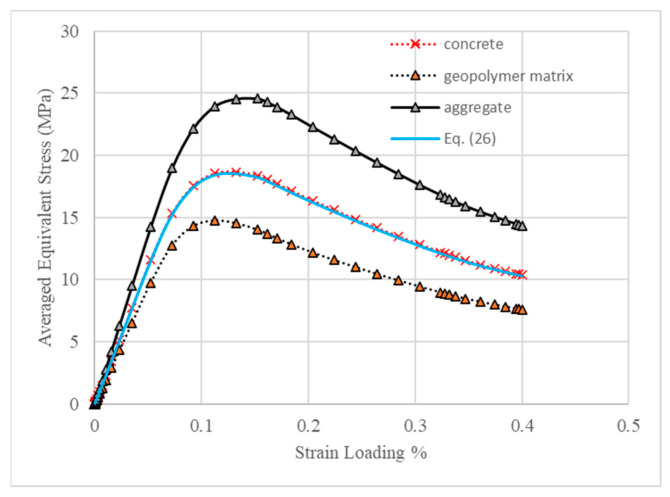
Evolution of averaged equivalent stress against strain loading of MesoModel_40PcAgg.

**Table 1 materials-18-00088-t001:** Elastic properties of concrete constituents.

Properties	Symbols	Mortar	Aggregate
Young’s modulus (MPa)	*E*	12,000	60,000
Poisson’s ratio	υ	0.2	0.2

**Table 2 materials-18-00088-t002:** Parameters for damage model of concrete constituents.

Properties	Symbols	Mortar	Aggregate
damage function parameters	$k_{0}$	0.77	0.78
	$k_{1}$	0.77	0.78
	$k_{2}$	0.15	0.12
damage threshold	$\gamma_{0}$	0.0001	0.0005
maximal degradation	α	0.9	0.9
rate of damage evolution	β	620	620
nonlocal range parameter	*c*	250	250

**Table 3 materials-18-00088-t003:** Elastic properties of different phases of the concrete.

Properties	Symbols	Mortar	Aggregate
Young’s modulus (MPa)	*E*	19,000	72,000
Poisson’s ratio	υ	0.2	0.2

**Table 4 materials-18-00088-t004:** Damage model parameters of different constituents of the concrete.

Properties	Symbols	Mortar	Aggregate
damage function parameters	$k_{0}$	0.73	0.78
	$k_{1}$	0.73	0.78
	$k_{2}$	0.25	0.12
damage threshold	$\gamma_{0}$	0.0001	0.0005
maximal degradation	α	0.9	0.9
rate of damage evolution	β	150	150
nonlocal range parameter	*c*	20	25

**Table 5 materials-18-00088-t005:** Elastic properties of geopolymer and granite.

Properties	Symbols	Geopolymer (Fly Ash)	Aggregate (Granite)
Young’s modulus (GPa)	*E*	14.7 [43]	52.05 [45]
Poisson’s ratio	υ	0.2 [43]	0.2 [45]

**Table 6 materials-18-00088-t006:** Damage model parameters of geopolymer and granite.

Properties	Symbols	Geopolymer	Aggregate
damage function parameters	$k_{0}$	0.72	0.78
	$k_{1}$	0.72	0.78
	$k_{2}$	0.29	0.12
damage threshold	$\gamma_{0}$	0.0001	0.0005
maximal degradation	α	0.9	0.9
rate of damage evolution	β	150	150
nonlocal range parameter	*c*	20	25

**Table 7 materials-18-00088-t007:** Compressive strength of geopolymer concrete.

Aggregate Volume Fraction %	Compressive Strength (MPa)	Increment %
20	14.2	-
30	16.6	17.0
40	19.6	37.9

**Table 8 materials-18-00088-t008:** Poisson’s ratio of geopolymer concrete.

Aggregate Volume Fraction %	Poisson’s Ratio
20	0.252
30	0.251
40	0.251

## Data Availability

The original contributions presented in this study are included in the article. Further inquiries can be directed to the corresponding author.

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
