# Peer review of "Mesoscale Modeling for Predicting Effective Properties and Damage Behavior of Geopolymer Concrete"

_materials, 2024, doi:10.3390/ma18010088_

Round 1

Reviewer 1 Report

Comments and Suggestions for Authors

The manuscript describes the study where the authors devised an easy and simple approach for generating concrete mesoscale models and characterizing the angular characteristics of aggregate particles.

The topic and scope suit the journal Materials, and the special issue “Modelling, Assessing and Controlling Deterioration Process of Reinforced Concrete Structures”; the methods are appropriate for achieving the objectives and the conclusions are supported. The results have been compared with literature and experiments, this is good.

The limitations and future perspectives of the study and method have not been discussed.

It is enlightening to see new computational methods being applied in reinforced concretes and given the adequate rigor demonstrated within this work, it should be considered for publication in Materials, after revision.

The whole manuscript can benefit from a thorough editing. English usage needs to be improved. Content organization should be reconsidered: a fraction of content in section 4 on simulation procedure for geopolymer concrete should be moved to Methods rather than presented as Results.

Perhaps all the equations should be referenced.

Amongst the three mesh sizes used, which is recommended for future application?

Please discuss how the results generated from 2D geometry can compare with those from 3D geometry.

Some clarification may be useful: there are many similar Tables, 1 and 3, 2 and 4. Please explain what each is for.  Also the Table titles should have more details of which is reference data which is determined results.

Fig. 14, which part is axial which is lateral strain?

Amongst the three aggregate loading, which is commonly used in practice?

Please discuss the effect of geopolymer aggregate bonding and interface surface area (interfacial stress zone) on poisson's ratio. Is it because of the assumed 0 thickness that “it was noticed that volume fraction had little effect on the Poisson’s ratio of geopolymer?” Once would think the larger volume in 2D geometry would mean larger interfacial stress zone surface area, and this somehow would affect the Poisson’s ratio.

There are many tables and figures, please condense and combine.

Please also discuss the computational error and statistical reliability of the proposed method, and estimate the error (precision/standard deviation) of each determined parameter.

Perhaps the considerations from Geometric, electronic and elastic properties of dental silver amalgam γ-(Ag3Sn), γ1-(Ag2Hg3), γ2-(Sn8Hg) phases, comparison of experiment and theory. Intermetallics 18.5 (2010): 756-760 can be referenced here “A default maximum strain amplitude was chosen to be (i) sufficiently low that the material stays within the linear elasticity regime (within the harmonic oscillator approximation). However, this must also be (ii) sufficiently high that the distortion generates dissimilar structures, energies and stresses which are significantly higher than computational error.”

Comments on the Quality of English Language

needs to be improved.

Author Response

Response to Reviewer 1 Comments

Summary

Response to comments

Comments 1: The manuscript describes the study where the authors devised an easy and simple approach for generating concrete mesoscale models and characterizing the angular characteristics of aggregate particles. The topic and scope suit the journal Materials, and the special issue “Modelling, Assessing and Controlling Deterioration Process of Reinforced Concrete Structures”; the methods are appropriate for achieving the objectives and the conclusions are supported. The results have been compared with literature and experiments; this is good.

Response 1: We would like to thank the reviewer for their positive feedback and for taking the time to carefully assess our manuscript. We appreciate your acknowledgment of the relevance of our study to the scope of the journal and the special issue, as well as your recognition that the methods are appropriate and the conclusions well-supported. We are also glad that you found the comparison of our results with literature and experiments to be a positive aspect of the manuscript.

Comments 2: The limitations and future perspectives of the study and method have not been discussed.

Response 2: Thank you for your valuable feedback. We appreciate your suggestion to discuss the limitations and future perspectives of our study and method. In the revised manuscript, we have now outlined the limitations of our approach in the conclusions. We believe these additions help provide a more comprehensive view of the study's scope and its potential for future developments.

Comments 3: It is enlightening to see new computational methods being applied in reinforced concretes and given the adequate rigor demonstrated within this work, it should be considered for publication in Materials, after revision.

Response 3: We would like to sincerely thank the reviewer for the positive feedback and for recognizing the value of our work. We are also glad that the computational methods applied in our study have been appreciated and that the rigor of our approach has been acknowledged. Besides, we are grateful for the opportunity to revise the manuscript and have addressed the reviewer’s suggestions and comments carefully to ensure the paper meets the journal’s standards for publication. Thank you again for your constructive feedback, and we look forward to submitting the revised version of our manuscript.

Comments 4: The whole manuscript can benefit from a thorough editing. English usage needs to be improved. Content organization should be reconsidered: a fraction of content in section 4 on simulation procedure for geopolymer concrete should be moved to Methods rather than presented as Results.

Response 4: We greatly appreciate the reviewer’s thoughtful feedback regarding the English usage and organization of the manuscript. We agree that improving language clarity is essential, and we have carefully revised the manuscript to enhance English usage and readability. Regarding the content organization, while we understand the reviewer’s suggestion to move the discussion of the aggregate gradation curves from Section 4 to the Methods section, we believe that the current organization helps maintain the flow of the paper and adds clarity to the findings. However, we are open to any further suggestions and are happy to make adjustments if necessary.

Comments 5: Perhaps all the equations should be referenced.

Response 5: Thank you for your helpful suggestion. We agree that referencing all the equations in the manuscript would improve clarity and ensure proper attribution to the methods and models used. We have now added appropriate references for all equations in the revised manuscript. We appreciate your attention to this detail and believe this revision enhances the overall readability of the paper.

Comments 6: Amongst the three mesh sizes used, which is recommended for future application?

Response 6: Thank you for your insightful question regarding the recommended mesh size for future applications. In the manuscript, we explored three different mesh sizes to evaluate their impact on model accuracy and computational efficiency. Based on our findings, we observed that variations between models with different mesh sizes were negligible and as the mesh went finer, compressive strength was much closer to that of experiment. Therefore, for future applications, we would recommend using the model with a mesh size of 0.5 mm. However, depending on the specific requirements of the application, the choice of mesh size can be adjusted accordingly.

Comments 7: Please discuss how the results generated from 2D geometry can compare with those from 3D geometry.

Response 7: Thank you for your thoughtful suggestion to compare the results from the 2D mesoscale model with those from a 3D mesoscale model. We would like to highlight that the results generated from the 2D mesoscale model in our study matched well with the experimental data, demonstrating the reliability of this approach for the scope of our current research. While we acknowledge that a 3D mesoscale model would offer a more detailed representation of the system, the use of a 3D model is beyond the scope of the current study. However, we recognize the value of a 3D model in capturing additional spatial and vertical dynamics that a 2D model cannot fully resolve. We plan to consider the implementation of a 3D mesoscale model in future research to build on the findings from this study and explore more complex interactions. Thank you again for your valuable input, which has helped further clarify the scope and direction of our work.

Comments 8: Some clarification may be useful: there are many similar Tables, 1 and 3, 2 and 4. Please explain what each is for. Also, the Table titles should have more details of which is reference data which is determined results.

Response 8: Thank you for your helpful comment regarding the clarity of the tables. We appreciate your observation about the similarities between Tables 1 and 3, and Tables 2 and 4. Tables 1 and 3 present the elastic properties of the constituents of different mix designs of concrete, while tables 2 and 4 contain the damage model parameters for the phases of different mix designs of concrete. Additionally, we have updated the table titles, making it clearer which data are associated with each specific aspect of the study. We hope these revisions address your concerns and improve the clarity of the manuscript. Thank you again for your constructive feedback.

Comments 9: Fig. 14, which part is axial which is lateral strain?

Response 9: Thank you for your valuable comment regarding the interpretation of strain components in Fig. 14. We apologize for any confusion caused by the presentation of axial and lateral strain. In the figure, the axial strain was plotted on the left side while the lateral strain was plotted on the right side. To improve clarity, we have updated the figure caption to explicitly mention the axial and lateral strain components, and we believe this will help avoid any further confusion. Thank you again for pointing this out, and we hope this clarification enhances the readability of the figure.

Comments 10: Amongst the three aggregate loading, which is commonly used in practice?

Response 10: Thank you for your thoughtful question regarding the commonly used aggregate volume fractions in practice. We would like to clarify that all three aggregate volume fractions investigated in our study can indeed be used in practice, depending on the specific requirements of the concrete mix and the desired properties. In practical applications, the choice of aggregate volume fraction is influenced by various factors, such as the required strength, workability, durability, and environmental conditions. Each of the three fractions can be selected based on these considerations. For example, a higher volume fraction may be chosen for applications requiring increased strength, while a lower fraction might be used for applications where workability or cost considerations are more important.

Comments 11: Please discuss the effect of geopolymer aggregate bonding and interface surface area (interfacial stress zone) on Poisson’s ratio. Is it because of the assumed 0 thickness that “it was noticed that volume fraction had little effect on the Poisson’s ratio of geopolymer?” Once would think the larger volume in 2D geometry would mean larger interfacial stress zone surface area, and this somehow would affect the Poisson’s ratio.

Response 11: Thank you for your insightful comment and suggestion to discuss the effect of geopolymer-aggregate bonding and the interfacial transition zone on Poisson’s ratio. We appreciate your thoughtful observation about the relationship between volume fraction and Poisson’s ratio in geopolymer systems. In our study, the Poisson’s ratio of geopolymer concrete was observed to be relatively insensitive to the volume fraction of aggregates. We believe this outcome arises primarily from the assumption of zero thickness for the interfacial transition zone in our model. This simplification results in the interface between the geopolymer matrix, and aggregates being treated as a perfect bond. We made this simplification due to lack of experimentally measured properties for the interfacial transition zone between geopolymer matrix and aggregate. When such data are available in future work, we plan to consider a more refined representation of the interfacial transition zone, including a nonzero thickness for the interfacial transition zone and varying bonding characteristics, to better capture these effects and their influence on the material's mechanical properties, including Poisson’s ratio. Thank you again for your valuable comment, which has helped clarify the limitations of the current model and opened up directions for future research.

Comments 12: There are many tables and figures, please condense and combine.

Response 12: Thank you for your valuable feedback. We appreciate your suggestion to condense and combine the tables and figures. After carefully reviewing the manuscript, we believe that each table and figure serves an important role in presenting the data clearly and effectively. Each one is necessary to fully convey the complexity of the results and ensure that the information is accessible to readers. We feel that the current number and arrangement of tables and figures provide the most logical structure for the study’s presentation and allow the reader to easily understand the relationships and key points. However, we are open to any further suggestions you may have regarding their organization or presentation.

Comments 13: Please also discuss the computational error and statistical reliability of the proposed method and estimate the error (precision/standard deviation) of each determined parameter.

Response 13: Thank you for your insightful comment regarding computational error, statistical reliability, and error estimation. We appreciate your suggestion to address these aspects in more detail. We observe that the main source of computational error in our study arises from the mesh size used in the simulations. As shown in our sensitivity analysis, finer mesh sizes lead to more accurate results, and the error tends to decrease as the mesh resolution increases. This result highlights the importance of mesh refinement for obtaining precise predictions, though it also increases computational cost. Regarding statistical reliability, we acknowledge that a detailed statistical analysis, including the estimation of standard deviations and precision of the determined parameters, was outside the scope of the current study. However, we recognize the importance of this aspect and plan to incorporate a comprehensive statistical analysis in future work to assess the variability and reliability of the model’s predictions more rigorously. Thank you once again for your valuable feedback, which has helped to further strengthen our study. We believe that these additions clarify the computational aspects of our method and outline potential directions for future research.

Comments 14: Perhaps the considerations from Geometric, electronic and elastic properties of dental silver amalgam γ-(Ag3Sn), γ1-(Ag2Hg3), γ2-(Sn8Hg) phases, comparison of experiment and theory. Intermetallics 18.5 (2010): 756-760 can be referenced here “A default maximum strain amplitude was chosen to be (i) sufficiently low that the material stays within the linear elasticity regime (within the harmonic oscillator approximation). However, this must also be (ii) sufficiently high that the distortion generates dissimilar structures, energies and stresses which are significantly higher than computational error.”

Response 14: Thank you for your comment regarding the reference to the study on dental silver amalgam and the considerations about strain amplitude. We agree that the properties of dental silver amalgam and geopolymer concrete are fundamentally different, and the specific considerations for strain amplitude in the context of the study on amalgam may not be directly applicable to our work. Thank you again for your valuable feedback.

Reviewer 2 Report

Comments and Suggestions for Authors

1. The authors should clearly indicate which FE software was used to perform the simulation.

2. The loading protocols such as loading rate or controlling mode, should be clearly indicated regarding compressive behaviors of geopolymer concrete.

3. Water/cement or water/binder ratio also plays a critical role in governing the material behaviors of concrete. Have the authors considered this issue and how was it considered?

Author Response

Response to Reviewer 2 Comments

Summary

Comments 1: The authors should clearly indicate which FE software was used to perform the simulation.

Response 1: Thank you for your valuable suggestion. We apologize for not specifying the finite element (FE) software used in the simulation earlier. In response to your comment, we have now clearly indicated the FE software used in the revised manuscript. The simulations were performed using ANSYS. We have updated the manuscript to ensure this detail is explicitly stated, and we believe it enhances the clarity and transparency of the methodology. Thank you again for bringing this to our attention.

Comments 2: The loading protocols such as loading rate or controlling mode, should be clearly indicated regarding compressive behaviours of geopolymer concrete.

Response 2: Thank you for your constructive comment regarding the loading protocols, particularly the loading rate and control mode, in the context of the compressive behaviour of geopolymer concrete. We agree that clarifying these details is important for the reproducibility and understanding of the study. In our simulation, we used displacement control to apply the compressive load. The displacement control mode ensures that the specimen is deformed at a constant rate, which is commonly used in materials testing to observe the full stress-strain behaviour up to failure without introducing rate-dependent effects. As mentioned in Section 2.3 of the manuscript, we have provided a detailed description of the testing protocols, including the loading procedure. These details should address any concerns regarding the control mode and loading conditions used in our study. We hope this clarification resolves your concern, and we appreciate your feedback, which has helped improve the manuscript.

Comments 3: Water/cement or water/binder ratio also plays a critical role in governing the material behaviours of concrete. Have the authors considered this issue and how was it considered?

Response 3: Thank you for your insightful comment regarding the water/cement or water/binder ratio and its influence on the material behaviour of concrete. We agree that the water/binder ratio is a critical factor in determining the mechanical properties of concrete, such as its strength and durability. However, in this study, the water/cement or water/binder ratio was not directly considered as a variable. Instead, the effect of the ratio was implicitly accounted for by using the properties of the individual constituents of the concrete, including the binder and aggregate materials, which were determined based on experimental data. These properties influence the overall behaviour of the geopolymer concrete in a manner like how the water/binder ratio would, as they reflect the material characteristics that govern the performance of the composite. In future research, we plan to explore the direct effect of varying the water/binder ratio on the geopolymer concrete's mechanical behaviour, as we acknowledge that this factor plays an important role in the material's performance. We hope this explanation clarifies the approach taken in our study, and we appreciate your helpful feedback.

Reviewer 3 Report

Comments and Suggestions for Authors

This paper studies the "Mesoscale Modelling for Predicting Effective Properties and Damage Behavior of Geopolymer Concrete",  which is an interesting topic, while there are still some problems that need to be addressed as follows:

1. Abstract lack of specific value to show the effect of properties and damage behavior of geopolymer concrete with the consideration of its heterogeneous characteristics by means of mesoscale models combined with the regularized Microplane damage model.

2. Some grammar mistakes need to be checked, for example, it should be "To overcome the limitation of the experimental method" in line 48,  and the introduction is not very sufficient, The discrete elements method is also a good way to simulate the performance of the heterogeneous characteristics like this paper "Identify the Micro-Parameters for Optimized Discrete Element Models of Granular Materials in Two Dimensions Using Hexagonal Close-Packed Structures".

3. How did the author calibrate the parameters used in this finite element mesh of the mesoscale model?

4. How to consider the contact between the cement and aggregate?

5. There are lot of content in the conclusions that is copied from the previous section, please improve it some grammar mistakes are in the conclusion, and a space needs to be deleted in line 464.

Author Response

Response to Reviewer 3 Comments

Summary

This paper studies the "Mesoscale Modelling for Predicting Effective Properties and Damage Behavior of Geopolymer Concrete”, which is an interesting topic, while there are still some problems that need to be addressed as follows:

Comments 1: Abstract lack of specific value to show the effect of properties and damage behavior of geopolymer concrete with the consideration of its heterogeneous characteristics by means of mesoscale models combined with the regularized Microplane damage model.

Response 1: Thank you for your comment regarding the abstract. We understand the importance of providing specific values to highlight predicted properties and damage behavior. However, as the values would vary depending on the specific mix design of the geopolymer concrete, we intentionally did not include specific numerical values in the abstract. The focus of this research is to demonstrate the capability of the proposed mesoscale modelling approach in predicting the properties and damage behaviour of geopolymer concrete, considering its heterogeneous characteristics. The goal is to showcase the model's versatility and potential for different mix designs rather than to report specific values that would change with each design. We hope this clarification helps, and we appreciate your suggestion, which has allowed us to further emphasize the purpose of the study in the abstract.

Comments 2: Some grammar mistakes need to be checked, for example, it should be "To overcome the limitation of the experimental method" in line 48, and the introduction is not very sufficient, The discrete elements method is also a good way to simulate the performance of the heterogeneous characteristics like this paper "Identify the Micro-Parameters for Optimized Discrete Element Models of Granular Materials in Two Dimensions Using Hexagonal Close-Packed Structures".

Response 2: Thank you for your helpful comments. We have carefully revised the manuscript to address the grammatical issues you pointed out, including correcting the phrase in line 48 to “To overcome the limitation of the experimental method.” Regarding the introduction, we acknowledge that the discrete element method (DEM) is another valuable approach for simulating heterogeneous materials, as highlighted in the paper “Identify the Micro-Parameters for Optimized Discrete Element Models of Granular Materials in Two Dimensions Using Hexagonal Close-Packed Structures.” However, the DEM is out of the scope of this research, which primarily focuses on finite element modelling to capture the mechanical behaviour of geopolymer concrete. Thank you again for your constructive feedback, which has helped improve the clarity and precision of the manuscript.

Comments 3: How did the author calibrate the parameters used in this finite element mesh of the mesoscale model?

Response 3: Thank you for your question. The parameters used in the finite element analysis of the mesoscale model were calibrated by comparing the model’s results with experimental, numerical, and theoretical data.

Comments 4: How to consider the contact between the cement and aggregate?

Response 4: Thank you for your comment regarding the consideration of the contact between the binder and aggregate in the geopolymer concrete. In our study, due to lack of experimental data for properties of interfacial transition zone (ITZ) in geopolymer concrete, the thickness of ITZ between the binder and aggregate was assumed to be zero, and perfectly bonding between the binder and aggregate was taken. The simplified approach allowed us to focus on the overall mechanical behaviour of the geopolymer concrete, but we acknowledge that a more detailed model that accounts for the interfacial transition zone between the binder and aggregate could provide further insights. This could be explored in future work by introducing an explicit interfacial transition zone with its own set of properties to better represent the contact effects at the binder-aggregate interface. Thank you again for your thoughtful comment, which helps us consider the potential for further refinement in our modelling approach.

Comments 5: There are lot of content in the conclusions that is copied from the previous section, please improve it some grammar mistakes are in the conclusion, and a space needs to be deleted in line 464.

Response 5: Thank you for your valuable feedback regarding the conclusion section. We appreciate your observation that some content in the conclusion is repeated from previous sections. In response, we have revised the conclusion to make it more concise. Additionally, we have carefully reviewed the conclusion for grammar issues and made the necessary corrections to improve clarity and readability. Specifically, we have addressed the identified grammar mistakes to ensure the text is polished and professional. Lastly, we have removed the extra space in line 464, as you pointed out, to maintain proper formatting throughout the manuscript. We hope these revisions meet your expectations and enhance the overall quality of the manuscript. Thank you again for your constructive comments.

Round 2

Reviewer 1 Report

Comments and Suggestions for Authors

Although the authors appear humble and appreciative in the "response" document by starting each of the 14 responses with "thank you" besides the overall document with another "thank you" statement, not much openess to constructive critisism nor willingness to address challenging issues has been demonstrated in the revision. Actually, the revised version does not differ much from the original version, seeing only a few insignificant red edits have been marked.

Please address major issues directly, such as evidence based comparison with 3D models to validate the 2D method proposed herein. The gratitude can be demonstrated in the revised manuscript by providing clarifications to queries. These are not for clarifying for the reviewers but for clarifying ambiguity in the manuscript.

This reviewer does not agree that detailed statistical analyses, including the estimation of standard deviations and precision of the determined parameters, are outside the scope of the study. Without these, the method can not be validated and no rational conclusion can be drawn.

Author Response

Comments 1: Please address major issues directly, such as evidence-based comparison with 3D models to validate the 2D method proposed herein. The gratitude can be demonstrated in the revised manuscript by providing clarifications to queries. These are not for clarifying for the reviewers but for clarifying ambiguity in the manuscript.

Response 1: Thank you so much for your suggestion of further validating the proposed 2D mesoscale modelling with 3D mesoscale models. In this research, we’ve validated the 2D mesoscale models using experimental data. We agree with the reviewer that 3D mesoscale models can reveal more spatial characteristics of both aggregate and concrete and do hope to extend our research to three dimensional aspects in future studies. We’ve updated the manuscript in the conclusions with the following explanation to show this limitation and further direction of our studies: “we would like to highlight that the results generated from the 2D mesoscale model in our study matched well with the experimental data, demonstrating the reliability of this approach for the scope of our current research. However, we recognize the value of a 3D model in capturing additional spatial and vertical dynamics that a 2D model cannot fully resolve. We plan to consider the implementation of a 3D mesoscale model in future research to build on the findings from this study and explore more complex interactions.”

Comments 2: This reviewer does not agree that detailed statistical analyses, including the estimation of standard deviations and precision of the determined parameters, are outside the scope of the study. Without these, the method cannot be validated, and no rational conclusion can be drawn.

Response 2: Thank you very much for your valuable feedback. We acknowledge that statistical analysis is very important for showing the robustness of the proposed method. In this research, more than 18 material parameters need to be determined. Although it’s not easy to directly calculate standard deviations and precisions of these parameters, we’ve updated our manuscript with a more understandable step-by-step explanation of how these parameters are calibrated:

  1. Experimental data: obtain stress-strain data from experiments, e.g., Nguyen et al., for concrete.
  2. Initial guess: set the initial empirical damage parameters, Young’s modulus, Poisson’s ratio and other material constants for both aggregate and mortar.
  3. Simulation: run an initial simulation of mesoscale modelling using the initial values.
  4. Error analysis: compare the FEA results with the experimental stress-strain curve.
  5. Adjust parameters: modify material constants.
  6. Iterate: repeat until the FEA curve matches the experimental data well.

Reviewer 2 Report

Comments and Suggestions for Authors

The reviewer has no more comments.

Author Response

Comments 1: The reviewer has no more comments.

Response 1: Thank you very much for taking the time to review this manuscript.

Reviewer 3 Report

Comments and Suggestions for Authors

The author has responded some of my comments, For this questions, How did the author calibrate the parameters used in this finite element mesh of the mesoscale model?

The author answer that The parameters used in the finite element analysis of the mesoscale model were calibrated by comparing the model’s results with experimental, numerical, and theoretical data. please make it more clear step by step in the manuscript since it is one of the most important content to support all the left analysis data content.

Comments on the Quality of English Language

The English could be improved to more clearly express the research.

Author Response

Comments 1: The author has responded some of my comments. For this question, how did the author calibrate the parameters used in this finite element mesh of the mesoscale model? The author answer that the parameters used in the finite element analysis of the mesoscale model were calibrated by comparing the model’s results with experimental, numerical, and theoretical data. please make it more clear step by step in the manuscript since it is one of the most important content to support all the left analysis data content.

Response 1: Thank you for your valuable feedback regarding the calibration of parameters used in the finite element analysis, and we apologized for not presenting the calibration process in a clear way. We’ve updated our manuscript with a more understandable step-by-step explanation:

  1. Experimental data: obtain stress-strain data from experiments, e.g., Nguyen et al., for concrete.
  2. Initial guess: set the initial empirical damage parameters, Young’s modulus, Poisson’s ratio and other material constants for both aggregate and mortar.
  3. Simulation: run an initial simulation of mesoscale modelling using the initial values.
  4. Error analysis: compare the FEA results with the experimental stress-strain curve.
  5. Adjust parameters: modify material constants.
  6. Iterate: repeat until the FEA curve matches the experimental data well.